# A Simulator to Determine the Evolution of Disparities in Food Consumption between Socio-Economic Groups: A Brazilian Case Study

**Pedro Gerber Machado [1],\*, Julia Tomei [2] , Adam Hawkes [1] and Celma de Oliveira Ribeiro [3]** 

[1]    Chemical Engineering Department, Imperial College London, London SW7 2AZ, UK; a.hawkes@imperial.ac.uk
[2]    Institute for Sustainable Resources, University College London, London WC1H 0NN, UK; j.tomei@ucl.ac.uk
[3]    Production Engineering, University of São Paulo, Sao Paulo 05508-010, Brazil; celma@usp.br
\*    Correspondence: p.gerber@imperial.ac.uk

**Abstract:** Food is a fundamental right that deserves attention but is usually dealt with from the supply side in aggregated models that use macroeconomic variables to forecast the demand and the required supply. This study challenges this paradigm by developing a simulator to analyze food consumption from the demand side and estimate the evolution of disparity in food consumption over time with respect to region, sex, ethnicity, education, and income. This novel approach was applied to Brazil using household expenditure surveys to feed serial neural networks. Results show that the 'poorer' north and northeast of Brazil encounter the lowest consumption of food and are therefore the most food vulnerable regions. This trend continues to 2040. The 'richer' south and southeast regions have higher food consumption, which varies according to sex, ethnicity, education, and income. Brazil has contrasting issues with some groups having considerably higher food consumption, while other groups still have less than the threshold for healthy consumption. Now, the country not only has to deal with the food access by the most vulnerable due to the latest economic declines but also to deal with excess consumption, the so-called "double burden of malnutrition".

**Keywords:** food consumption; sustainable development goals; inequality; Brazil; neural network

## 1. Introduction

Since the 1960s, global food production has more than doubled [1]. This has been driven by a rapid increase in the global population, average daily per capita calories consumed (from 2196 kcal/capita/day in 1961 to 2874 kcal/capita/day in 2013 [2]), and the result of higher living standards [3–5].

Until 2015, reductions in inequalities in food have been the result of several global initiatives that have sought to end hunger [1]. For example, Target 1C of the Millennium Development Goals (MDGs), which ran from 2000–2015, aimed to halve the proportion of people who suffered from hunger. Brazil made good progress on this target, reducing the proportion of the population suffering from hunger from 14.8% in 1990 (or 22.6 million people) to less than 5% in 2015 [6,7].

Building on the successes and failures of the MDGs, in 2015 the United Nations adopted the 2030 Agenda for Sustainable Development, which is underpinned by 17 Sustainable Development Goals (SDGs). Of these, SDG2 aims to achieve zero hunger by 2030 by ensuring access for all to safe, nutritious, and sufficient food all year round [8]. Although Brazil continued to face food inequalities in 2009, with 35.5% of Brazilian households experiencing some form of food insecurity [9], the country was moving towards the achievement of SDG2, reducing the prevalence of undernourished people in the population [10,11].

With a population close to 210 million, Brazil is a continent-sized country and is characterized by geographical, climatic, ethnic, and cultural diversity. It is also a highly unequal country with significant income disparities across its regions, as well as between rural and urban areas. In recognition of Brazil's inequalities, its constitution brings legal ground to redress this legacy and repay the accumulated 'social debt' [12]. Until 2015, significant social entitlements were set out in legislation, while various federal assistance programs, such as the "Bolsa Familia", had been created to provide financial aid to the poorest [12,13]. However, these do not go far enough. If Brazil is to achieve the SDGs by 2030, substantial progress in addressing the country's inequalities will need to be made, especially after following the global trend of increase in inequality perceived in the last five years [1,12]. Furthermore, with the 2020 Covid-19 pandemic, there is an increased likelihood that the world, including Brazil, will not reach the SDG targets, with possible increases in poverty and undernourishment [14–16]. The Food and Agriculture Organization (FAO) stresses that economic slowdowns and, as is the case of Brazil, downturns have contributed to the recent rise in hunger and more targeted efforts to address causes of hunger and malnutrition, such as inequality and poverty, are necessary to achieve SDG2 [1].

Even with such intricate inherent connections between SDG2 and other SDGs [17], the literature has focused on food demand from the supply side and has not yet addressed how food consumption relates to socio-economic disparities and regional inequalities. To fill this gap, this study aimed to provide evidence to support the challenge of reaching the SDGs by developing a model to analyze the future of food consumption. It focuses on Brazil, a country which has experienced a reduction in food inequalities due to strong policy measures. A model based on neural networks is developed aiming to identify how the regional food consumption evolves over time. The proposed simulator for future food consumption does not assume a predetermined temporal analytical structure, as is done, for example, in Bodirsky et al. [18], and goes beyond the use of macroeconomic variables, such as GDP [19,20] or GDP per capita [21], which can be limiting when it comes to understanding inequality and disparities among different socio-economic groups, since aggregated data does not allow the comparison of vulnerable groups within a country, where vulnerability is considered as the inability to cope with food insecurity [22]. Furthermore, with such disaggregation is possible to address what the World Health Organization (WHO) calls the "double burden of malnutrition", which is characterized by the coexistence of undernutrition and overweight and obesity, a phenomenon prevalent in countries that experience high consumption inequality [23,24].

The model uses characteristics of the household reference person (HRP) to simulate food consumption based on sex, ethnicity, income, and education using Brazilian household expenditure surveys data. It aims to better direct public policy regarding food inequality and disparities by enabling more targeted direction of policy instruments due to the disaggregation. The paper analyses differences by region and explores actions needed to address disparities related to sex, ethnicity, income, and education, drawing from the SDGs to develop national guidelines. With slowing and retracting economies, such as that seen in Brazil, it is critical to understand future trends in hunger and malnutrition, especially given the latest underwhelming projections of the global economy.

## 2. Literature Review

### 2.1. Previous Models

Given the importance of projecting food demand to any country, a wide variety of models are available to project demand, which have different complexities and methodologies. While providing sufficient food for all people worldwide is a vital goal, the production of food has many negative environmental impacts. As a result, there is a need for models that predict future food demand to support the design of the agricultural, economic, and conservation policies required to feed over 9 billion people in the world by 2050 with minimal harm to the environment [21,25]. Indeed, several of the models that have been created and used to determine future food demand at global, national, and regional scales have a focus on reducing environmental impacts and the sustainable supply of food.

A first set of models use a linearized version of the Engel's curve to project food demand. For example, Olowa et al. [20] treat food demand as a linearized Engel's curve (logarithm of demand equal a constant plus the logarithm of income, most of the times GDP). Pardey et al. [26] also use Engel's model and include other variables related to gender and regional differences. Another type of model used for demand projection encompasses partial and general equilibrium models [27], which try to explain the behavior of supply, demand, and prices in a whole economy based on neoclassical economics [28,29]. A third type of model commonly used in food projections is time series; these are used by the Brazilian Ministry of Agriculture and Livestock to project food consumption and production [19]. Time-series models are mathematical forecasting models, which seek to find the dependence of the future value on past values [30,31].

Moreover, in their review of food demand forecasting models, Flies et al. [25] show that 17 out of 22 articles that dealt with food demand focused on the impacts on land use, climate change, and agricultural supply, rather than issues related to poverty, food access, and food consumption disparities within or between countries. From the same 22 articles, only five focused on issues such as human health, hunger, food security, and nutrition. Furthermore, Flies et al. [25] reveal that 21 out of 22 papers use either GDP alone (2 papers out of 22) as the main determinant of per capita food demand or in combination with prices (15 papers out of 22) from partial or general equilibrium models, with fewer including other variables such as income inequality/disparity (2 papers out of 22), population age structure (1 paper out of 22), and time (2 papers out of 22).

Two important examples of models dealing with more than GDP and focus on issues related to hunger and food access are Kii et al. [32] and Bijl et al. [21]. Kii et al. [32] estimate the Ginicoefficient for food consumption within countries based on FAO data. The authors find that a reduction in food access disparity will reduce the total food demand significantly in developing countries, leading to a smaller required harvesting area. Bijl et al. [21] also focus on the relationship between income and dietary patterns, with income inequality separated from the inequality of food distribution. The authors project that the number of undernourished people declines in most regions by 2100, but they also recognize that the degree to which the variance of consumption will decrease autonomously is highly uncertain.

Even though food consumption models have been developed before in seemly exhaustive ways, none of them have made use of modern neural networks. Neural networks are powerful models with a universal approximation property [33], which allows for more accurate representations of interactions between variables. Its applications range from object detection or face recognition [33] to finance and economic forecasting [34] and the detection of the probability of low-income households falling into fuel poverty [35]. In this study, neural networks are innovatively applied to the issue of food consumption and intra-country disparities.

### 2.2. Food Consumption: Availability Versus Intake

Food consumption can be represented by food availability or food intake. Food availability is used for macroeconomic planning and projections, while food intake provides information on lifestyle and the health of a population regarding noncommunicable diseases, such as obesity [36]. Food availability is often related to the supply of food, and FAO defines it as the primary commodity and processed commodities potentially available for human consumption [37]. On the other hand, food consumption from the intake perspective sees food as energy, and only what is ingested is considered from a dietary and nutritional point of view [38,39].

At the household level, food consumption can be calculated using Food Frequency Questionnaires (FFQ). The questionnaires record the amount of food bought for consumption and the total calories are calculated through the energy content of each food item. The Brazilian Household Expenditure Survey (HES) is an example of an FFQ. A second approach estimates food intake, here called the Food Intake Approach (FIA). A sample of the population is taken, and, over a period of time, normally 24 h, everything that each individual eats is recorded. In general, the FIA uses images of food and leftovers, and the content of the pictures are analyzed using computer processing [38]. De Souza et al. [40]

further explain that FFQ values are expected to be higher than values obtained from FIA, because it does not include consumption-level waste (i.e., that wasted at retail, restaurant, and household levels) and therefore represents food available for consumption at the retail level, rather than actual food intake [41].

In this study, consumption is based on the Brazilian HES; therefore, it is seen through the availability perspective, not intake, and should be seen as the food supply available for human consumption, i.e., food available at the retail level.

## 2.3. Defining Inequality

The problem of inequality is of central importance in policymaking of national governments and international agencies. There is no single accepted definition of 'inequality' nor is there a common political justification regarding the need for its elimination [42]. Equality can be understood as equality in the minimum thresholds that allow the exercise of citizenship; in other words, equal rights serve as the basis for social pacts that reflect more opportunities for those who have the least [42]. Furthermore, a variety of economic and political considerations diverge in theorizing the role and importance of equality and, conversely, of inequality [43]. Equality is often discussed in terms of incomes, opportunities, wealth, achievements, rights, or other factors. The multidimensional nature and the complexity of defining and measuring inequalities also add to the multitude of uses for the term inequality. Inequality can be measured on multiple scales of analysis, depending on the space in which different individuals, households, or countries, are to be compared [43].

Concerning economic inequality, two views are most important. One is an achievement-oriented perspective and deals with inequality of well-being that may be the result of talent and effort and is independent of circumstances, such as ethnicity, gender, family, etc. The second view is concerned with the inequality of opportunities, that is, it focuses only in the circumstances beyond one's control. Studies only started to focus on inequality once it became apparent that income inequality was detrimental to economic growth. Before that, distributional concerns were mostly put aside, as growth was thought to eventually reach every layer of society [44].

Additionally, the problem of measuring inequality in even more complex. The simplistic idea that inequality is the scalar difference between two groups does not represent the interpersonal differences in level of material wealth within a given population [45]. For that reason, inequality in this study is used when referring to the overall concept of inequality of opportunities, and, when comparing two groups within a population, the term disparity is used instead.

## 3. An Introduction to Brazil

### 3.1. Brazilian Regional Differences

Brazil is a country of continental proportions and covers 8.5 million km$^2$. It is divided into five regions (shown in Figure S7 shows the relationship between neurons and the regression R between targets and outputs) [46], which vary greatly in terms of their population, economic activities, and levels of development.

Historically, the country has been divided between the more prosperous South (S) and Southeast (SE) and the less developed North (N) and Northeast (NE). In the center of the country rests its Capital, Brasília, in the Central-west (CW) region (Figure 1). The SE and S regions, which have around 50% of the population, together generate around 72% of Brazil's GDP. By contrast, the NE has around 25% of the population and accounts for just 13% of GDP [47,48]. While there are signs of convergence between the regions, in 2013 moderate poverty was 3.3% in the SE compared to 17.6% in the NE [12]. According to the World Bank [12], more than half of the 7.6 million Brazilians living in extreme poverty are located in the NE.

Although Brazil's GINI index fell from 0.594 in 2001 to 0.513 in 2015 [12,49], it has recently seen an increase, reaching 0.533 in 2017 [50]. The N and NE regions present the highest income inequalities, and, in 2010, 3 of the 5 worst state GINI indices in the country were in those regions.

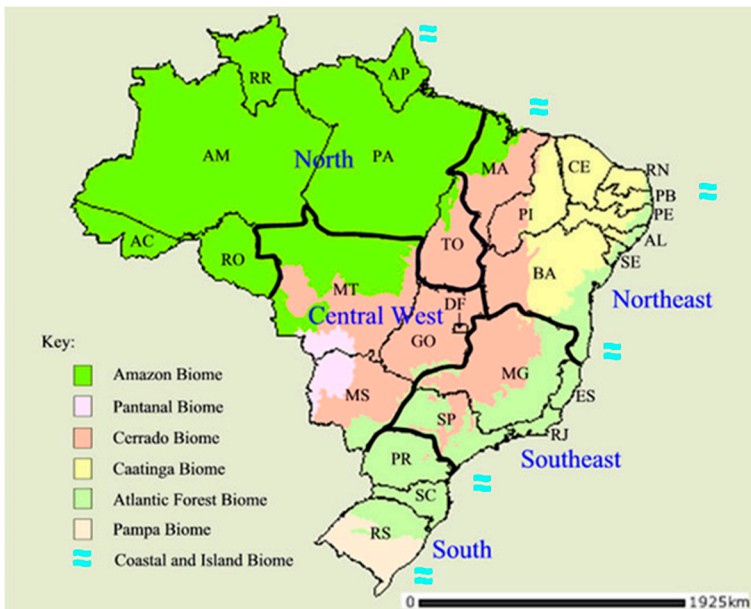

**Figure 1.** Brazil and its regions. Source: Hargreaves [51].

When analyzing income by ethnicity and sex, the country's inequalities become apparent. Income varies according to region, ethnicity, and sex, with the highest income belonging to Caucasian males living in the CW (mostly due to the country's capital and its high salaries in the public sector) and the SE (Figure 2). At the other end of the spectrum, the lowest incomes are earned by African-Brazilian females living in the NE and N. On average, Caucasian males in CW earn 2.75 times more than African-Brazilian females in the NE [49].

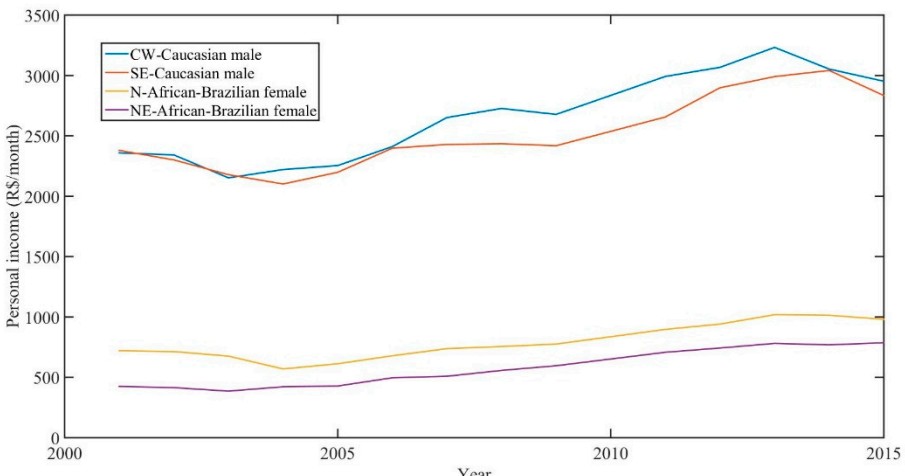

**Figure 2.** Personal income for selected groups (Caucasian male in Central-west (CW) and Southeast (SE) and African-Brazilian female in the Northeast (NE) and the North (N)) from 2001 to 2015 (in R$ of 2015). Source: IPEA [49].

### 3.2. Diets and Food Consumption in Brazil

Brazil is characterized by striking cultural hegemony when it comes to food, with rice, beans and meat being at the center of the majority of households' meals [52,53]. Each region has historically had

its own cuisine which reflects both environmental factors (e.g., climate, soil type, fauna) and individual histories of colonization and migration [54]. Their cuisine reflect their history, geography and climate (Brazil is located between the equator and more temperate climates, which facilitates the cultivation of a vast range of fruits and vegetables) and influence by Portuguese, Indigenous, African, other European, and—more recently—Asian cultures [54]. However, since the 1990s, the dietary patterns of Brazilians have been converging and homogenizing across the regions [54–57]. Considering its size and the diversity of available foods, the quality and quantity of food consumption in the country is diverse. In this context, income, prices, preferences, beliefs, and cultural traditions, as well as geographical, environmental, social, and economic factors, influence dietary consumption patterns [58].

Regardless of the approach used for determining food consumption, there are considerable differences in food consumption in Brazil [59]. For example, de Souza et al. [40] estimate the daily average food intake in SE Brazil by men to be 2188 kcal/day and for women 1570 kcal/day. The authors highlight the differences between their results, which used FIA, and other studies that use FFQ. For instance, Bonomo et al. [60] found an average intake of 2807 kcal/day for women and 3775 kcal/day for men, resulting in a difference of 70% between approaches.

Several other studies have used FFQ and FIA to analyze dietary information and the evolution in dietary quality in Brazil [4,61–63]. These studies show that Brazil has undergone rapid change, particularly through the uptake of processed foods with high calorie content and drinks with added sugar [61]. When it comes to a healthy diet, the Brazilian population is in the category of "needs improvement", i.e., it is characterized by low consumption of fresh fruit and vegetables, and high fat intake, particularly saturated fats [63]. In the NE, for example, consumption of regional foods is low, with higher intakes of processed ready-to-eat meals, which demand less time to prepare than regional foods [62]. Conversely, the N region has a lower prevalence of sugary soft drinks and higher presence of fish and is marked by the presence of manioc flour in meals [61]. These regional differences are, however, becoming less visible over time [61].

The recent changes in Brazilian diet lead to positive energy (calories) gains, which is a cause of the increase in obesity over the last 30 years [41]. Souza et al. [40], for example, found a low prevalence of low weight and high prevalence of pre-obesity/obesity in both men and women (49 and 45%). The authors also revealed a positive relationship between income and both the availability of food in the home and individual intake. This finding is supported by Moratoya et al. [53], who found that those on higher incomes were at greater risk from obesity. Pereda and Alves [56] also found a relationship between obesity and income with lower income households showing higher levels of being underweight, while obesity rates were higher in high income families. Income is also linked to the risk of hypertension due to dietary patterns [56].

Many authors highlight the need for more research on the impacts of diet on diverse societal groups, as well as across Brazilian regions [59,62,63]. Dos Anjos et al. [41] found a lack of data from the application of FIA covering the whole country. The authors found data on food availability in the HES of 1974/1975, 1987/1988, 1995/1996, 2002/2003, and 2008/2009, with only 2002/2003 and 2008/2009 covering all regions of the country [9,64]. Data on food intake, however, is only available in three of the surveys conducted over the last 30 years (1974/1975, 1989, and 2002). Further, there is an unequal distribution of studies and data in Brazil in the area of food and nutrition, with the N and NE regions being the most poorly studied [59]. Scholars have also argued for more studies that focus on other Brazilian regions to provide a comparison [63] and to determine how Brazilian regional foods can compose a healthy diet [62].

*3.3. Food and Nutritional Security Policies in Brazil*

Brazil has a long history of food and nutrition public policies, which were initiated in the mid-1930s in recognition of the profound economic, political, social, and cultural inequalities [65,66]. More recently, in recognition of the relationship between poverty, inequality and hunger, Brazil established a national plan for Food and Nutrition Security (FNS) [66]. The FNS created several short, medium and long-term

programs to fight hunger and poverty; these include *Programa Fome Zero* (PFZ, Zero Hunger Program), *Programa Nacional de Alimentação Escolar* (PNAE, National School Nutrition Program), and *Programa de Aquisição de Alimentos* (PAA, Food Acquisition Program), as well as income transfer programs, such as *Programa Bolsa Família* (PBF, Family Grant Program), which consolidated several smaller income transfer programs to increase the access to food and the purchasing capacity of low-income households [13,66]. Additionally, the 2006 food security law (LOSAN) provided a framework for the National Food and Nutrition Security System (Sistema Nacional de Segurança Alimentar e Nutricional—SISAN). The framework had the explicit aim to structure, coordinate and monitor public policies to ensure the right to adequate food in a decentralized way. The PNSAN (National Policy for Food and Nutritional Security) was approved in 2010, with clearly defined guidelines, procedures and funding mechanisms, monitoring, and evaluation of state actions on food and nutrition [6,11,13].

Through these programs, laws and frameworks, the proportion of households lacking sufficient food fell from 46.8% in 2002 [64] to 35.5% in 2009 [9]. Further, the proportion of people living in hunger was reduced by one-third [67], and the number of people living in poverty decreased by 20 million [67].

The increase in personal income between 2002 and 2009 was also a key factor for reducing the prevalence of undernourished people. A 25% increase in real incomes over that period [68] helped to reduce the prevalence of undernourished people in the population from 10.6% in 2002 to less than 2.5% in 2010 [10]. This decrease of undernourished and the increase in income of the period translated directly into a fast increase of average daily average food consumption from 2900 kcal/capita/day to 3200 kcal/day, as shown in Figure 3. Furthermore, results from time series models [69] based on the data in Figure 3. have shown that food consumption during this period was higher than it had historically been. One factor was likely to be government investments in programs to eradicate hunger and redistribute income, as well as increases in incomes, over that period. This is also highlighted by FAO, et al. [10], which show that sustained growth in GDP per capita in Latin America between the 1990s and mid-2010s was accompanied by significant reductions in poverty and undernourishment. Data from the FAO also show an important increase in the average national food consumption between 2002 and 2010, with a considerable jump from 2002 to 2004.

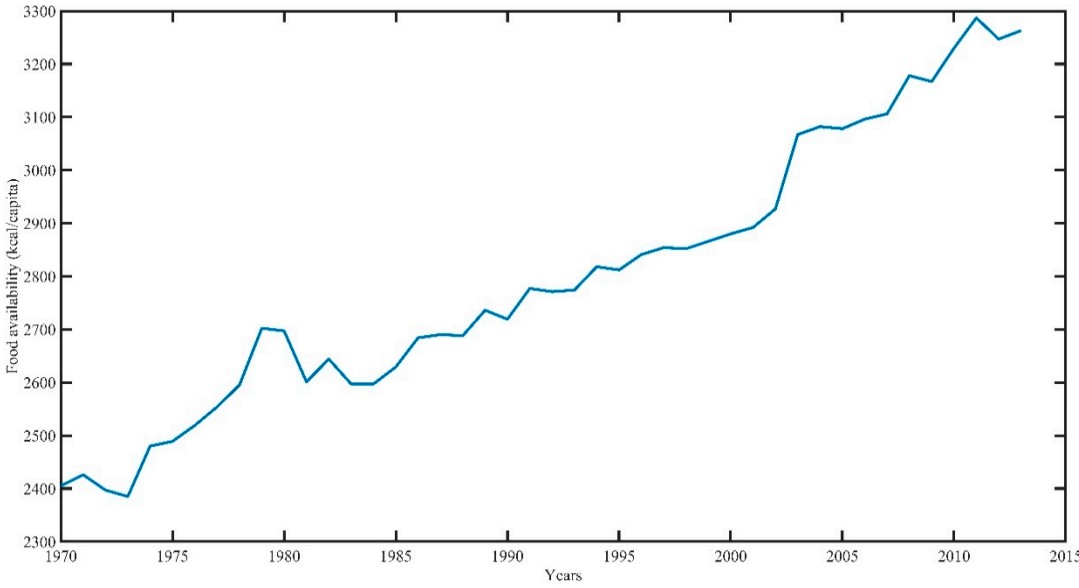

**Figure 3.** Average food consumption per capita (kcal/capita) from 1970 to 2013. Source: IPEA [2].

In 2016, the government at the time launched the second PNSAN, prepared by the Interministerial Chamber of Food and Nutrition Security (CAISAN), together with the National Council for Food and Nutrition Security (CONSEA), based on the deliberations of the V National Conference on Food and Nutrition Security, in an attempt to hold on to the previous success of the national programs against

inequality. These three bodies make up the governance of the Food and Nutrition Security agenda in Brazil [70].

However, with the election of president Jair Bolsonaro, CONSEA was dissolved, threatening SISAN and the PNSAN [71,72]. The end of the council has particularly concerning effects in this time of fiscal austerity measures, which could significantly affect the supply of the public health system's services, contributing to the deterioration of people's health and the increase of noncommunicable diseases such as obesity [71,73]. Furthermore, the recent cuts in governmental expenditures for Bolsa Familia [69] and the public budget limitations established for the next 20 years [74] put in jeopardy the gains made with the implementation of the measures in the PNSAN, worsening indicators, such as rising child mortality rates, increases in poverty, and unemployment rates going from 7% in 2012 to 12% in 2019, with signs that Brazil will return to the Hunger Map [1,71,75]. In the most recent era of economic slowdown, it appears that poverty is once more increasing [1]. Furthermore, the Covid-19 pandemic constitutes the biggest threat to the SDG of ending poverty (SDG1). Sumner et al. [14] estimate an increase in poverty, which, in some cases, represent the loss of approximately a decade in global progress in reducing poverty.

Having summarized Brazilian regional differences and inequalities, existing studies on food consumption patterns and quality in Brazil and the recent food policies in place, Section 4 describes the methodology used to project food consumption using household characteristics, serving as basis for the analysis of the projection of its inequality in time, with specific interest in the nature of inequalities related to region, sex, ethnicity, and education.

## 4. Materials and Methods

### 4.1. Rationale

Food consumption (the dependent variable) in this study is treated as non-linear function of five qualitative variables (predictors): region, sex, ethnicity, level of education, and income group (InG). The proposed simulator for future food consumption does not assume a predetermined temporal analytical structure for the non-linearity relationship between predictors and dependent variable, as is done, for example, in Bodirsky et al. [18], who define a "conversion" function based on the assumption that deviations of observed values from their proposed regression function gradually disappear. Therefore, to avoid predefined assumptions and to capture this non-linearity, an artificial intelligence model, in this case a neural network, is used as basis for the simulator.

Neural networks are universal function approximations [76] and are inspired and designed based on the neural structure of the human brain. Like the human brain, neurons are trained with the known record by comparing their classification to learn the real time environment. In order to avoid faulty learning, the error from the previous iteration is fed back into the network, and the weights are optimized. Neural networks are composed of an input layer, an output layer, and hidden layers (IL, OL, and HL in Figure 4, respectively). The predictors constitute the input layer which is the first layer while the output layer is the final layer containing output nodes for each dependent variable. The hidden layer is present between input and output layer and is made of N neurons (*n* in Figure 4) [77]. Equation (1) shows the structure of the model:

$$F_{con} = f_a(Reg, Sex, Eth, Edu, InG), \tag{1}$$

where $F_{con}$ is the food consumption of the household, $f_a$ is the representation of the non-linear relationship (neural network), *Reg* is the household region, *Sex* is the sex of the HRP, *Eth* is the ethnicity, *Edu* is the education level, and *InG* is the income group of the HRP.

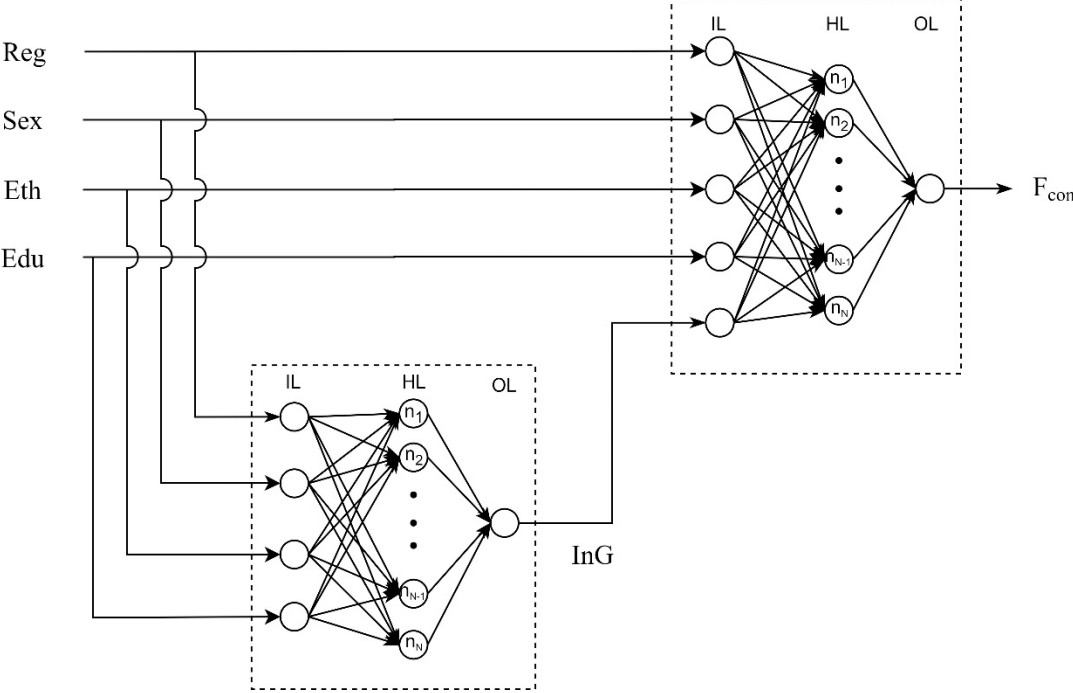

**Figure 4.** Rationale of the proposed simulator.

Food consumption dependence on these variables is determined by training a neural network model using historical data; however, the projection of food consumption still depends on the projection of such demographic indicators. While there is a considerable time-related evolution of ethnicity, sex, and education by region (increase in the female population, increase in the share of mixed-race, and increase in education level) [49,68,78], income does not follow the same time-related increase. Brazil has one of the highest income inequalities in the world [50], and it is well known that Brazilian household income varies according to sex, ethnicity, and housing region [49]. For that reason, a second non-linear function is defined for the income group to which a certain household belongs to, where it depends non-linearly on these predictors, as shown in Equation (2):

$$InG = f_b(Reg, Sex, Eth, Edu),\qquad(2)$$

where *InG* is the income group, $f_b$ is the non-linear function that defines the relationship between dependent variable and predictors, *Reg* is the household region, *Sex* is the sex of the HRP, *Eth* is the ethnicity, and *Edu* is the education level of the HRP.

Again, for the income group, a neural network is trained in order to capture the non-linearity among variables, without any predefined temporal analytical structure to determine the evolution of the dependent variable. Ultimately, the simulator can be represented as the diagram depicted in Figure 4. With the qualitative information of households for each year in the projection horizon (2020–2040), the simulator can estimate the food consumption.

The variables in the simulators are further explained in Table 1. Each of the variables will then be projected in terms of shares of total population to feed the simulator and estimate the yearly averages shown in the results in terms of sex, ethnicity, education, and income. Section 3.3. explains further the projection of variables for the 2040 horizon.

**Table 1.** Variables, their types, and classes (units).

| Type of Variable | Variable | Classes (Categorical) |
|---|---|---|
| | | Units (Continuous) |
| Categorical predictors | Sex—Sex of the person of reference of the household | Male<br>Female |
| | Eth—Ethnicity of the household reference person | Caucasian<br>African-Brazilian<br>Asian-Brazilian<br>Mixed heritage<br>Indigenous<br>Does not know |
| | Edu—Level of education of the household reference person | No schooling<br>4th grade<br>8th grade<br>High school<br>Undergraduate<br>Postgraduate |
| | InG—Group of income to which the household reference person belongs to | less than US$ 125/month/capita<br>US$ 125.01 to US$ 250/month/capita<br>US$ 250.01 to US$ 375/month/capita<br>US$ 375.01 to US$ 500/month/capita<br>and more than US$ 500/month/capita |
| Dependent | $F_{con}$—Food consumption per household per capita per day | kcal/household/capita/day (continuous) |

Section 4.2 will explain the implementation of the simulator, which required complex data manipulation to be able to provide a robust working model for the estimation of food consumption and to provide comprehensive information on the evolution of inequalities within the Brazilian food system.

*4.2. Implementation*

To build the household food consumption simulator, historical data was needed to train the corresponding neural networks. However, as mentioned in Section 3.2, the most recent data on physical consumption considering consumption inside and outside the household is from 2002. To comprehensively estimate physical consumption per household for the HES of 2008 and to maintain coherence among the two HES, a new method had to be developed and applied to both HES.

4.2.1. Estimating the Total Regional Household Per Capita Consumption per Day in Physical Quantities

For food consumption, the Brazilian HES are divided into: (i) consumption inside the home, which provides the total quantities in kilograms per household; and (ii) consumption outside the home, which includes only the monetary expenditure on food. Figure S1 through Figure S4 of the supplementary material show information on the Brazilian HES and FAO data used in this study.

In order to calculate the total consumption of food per capita in kilocalories ($KcalCapitaTot_{i,y}$), it was necessary to estimate food consumption outside the home (see Figure 5). First, the total kilocalorie consumed ($KcalTot_y$) was provided by multiplying consumption per capita (kcal/capita/year), obtained by FAO [2] for 2002 and 2008, by the total population in the same years.

From the 2002 and 2008 HES, the total consumption at home was calculated according to:

$$KcalHouseTot_y = \sum_i^{House_y} KcalHouse_{i,y}, \tag{3}$$

where $KcalHouseTot_y$ is the total consumption in kilocalorie in year *y*, $KcalHouse_{i,y}$ is the total consumption by household *i*, and $House_y$ is the total number of households in Brazil in year y (2002 or 2008).

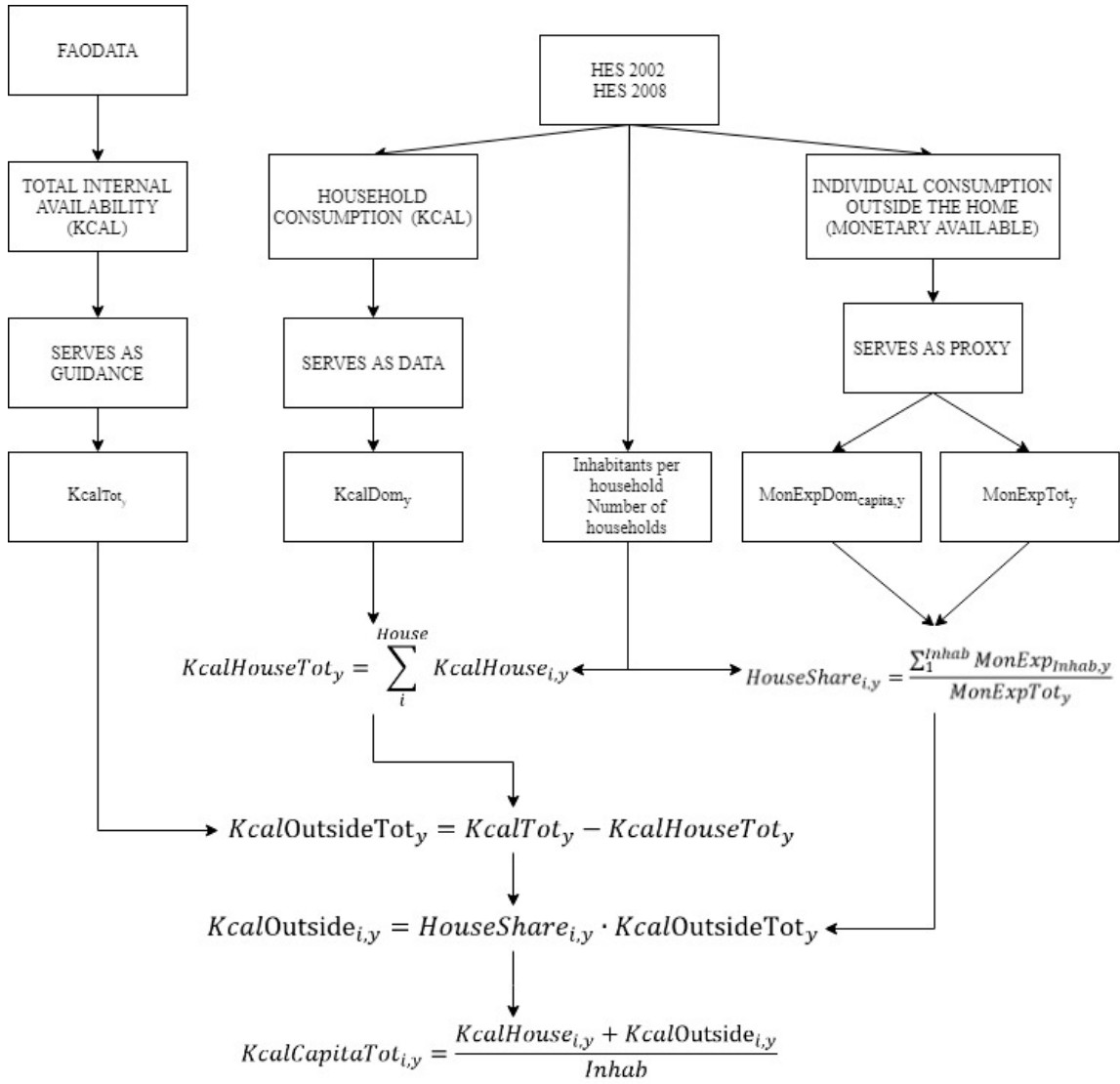

**Figure 5.** Method to estimate the total per capita consumption (kcal/capita/year).

With both values, it was possible to estimate what is consumed outside the home (restaurants, diners, etc.) using:

$$KcalOutsideTot_y = KcalTot_y - KcalHouseTot_y, \tag{4}$$

where $KcalOutsideTot_y$ is the total amount consumed outside the household in the country in year $y$, $KcalTot_y$ is the total consumed in Brazil in year $y$, and $KcalHouseTot_y$ is the total consumption inside the household in year $y$.

To estimate the consumption by each household, Equation (5) is used:

$$KcalOutside_{i,y} = HouseShare_{i,y} \cdot KcalOutsideTot_y, \tag{5}$$

where $KcalOutside_{i,y}$ is the consumption in each household $i$, in year $y$, $KcalOutsideTot_y$ is the total consumption in the country in year, and $HouseShare_{i,y}$ is the share of total monetary expenditures in outside the home food consumption in year $y$ by individuals in household $i$, calculated as shown in Equation (6):

$$HouseShare_{i,y} = \frac{\sum_1^{Inhab} MonExp_{j,y}}{MonExpTot_y}, \tag{6}$$

where $HouseShare_{i,y}$ is the share of total monetary expenditure outside the home on food consumption in year $y$ by individuals in household $i$, $MonExp_{j,y}$ is the monetary expenditure in food products by inhabitant $j$ in year $y$, $Inhab$ is the number of inhabitants in the household $i$, and $MonExpTot_y$ is the total monetary expenditure in Brazil in year $y$.

Finally, the total (outside and inside the household) per capita food consumption in household $i$ and year y is calculated with Equation (7):

$$KcalCapitaTot_{i,y} = \frac{KcalHouse_{i,y} + KcalOutside_{i,y}}{Inhab_i},$$ (7)

where $KcalCapitaTot_{i,y}$ is the final kilocalorie consumption per capita in household $i$ in year $y$, $KcalHouse_{i,y}$ is the kilocalorie consumption inside the home, $Kcal$Outside$_{i,y}$ is the consumption outside the home, and $Inhab_i$ is the number of inhabitants in household $i$.

The resulting averages for each household are then clustered, as discussed below. The average consumption per region is shown in Tables S7 and S8 of the Supplementary Materials.

### 4.2.2. Clustering Households

Due to the high uncertainty and the large datasets obtained from both HES (2002 and 2008), households were clustered by region and by year to decrease the variability in the data. There are numerous ways to sort cases into groups and the choice of a method depends on, among other things, the size of the data file. Since this study deals with a large amount of data (over 4000 cases per region) with categorical variables, a two-step procedure [79] was used based on the variables in Table 1. (income group, sex, ethnicity, education, and food consumption). Figure S5 in the Supplementary Materials shows the resulting clusters in terms of income and consumption.

### 4.2.3. Neural Networks

To train this first neural network presented in Figure 4, the variables used as input were sex, ethnicity, and education of the HRP, found in Table 1. The output of the neural network was one of five income groups.

The resulting income of each household was then fed into the second neural network, along with the other characterizing variables, allowing for the estimation of the food consumption in that household.

The first neural network is a pattern recognition feedforward back propagation network, trained with a scaled conjugate gradient backpropagation algorithm. The number of neurons in the hidden layer and the number of hidden layers were defined based on training the networks repeatedly until the error was minimized. Figure S6 in the Supplementary Materials shows the error of the trained neural network to simulate household income.

The second neural network is an input-output fitting feedforward back propagation network, trained with a Levenberg-Marquardt backpropagation algorithm. The simulation of food consumption also required an iteration to minimize the error. Figure S7 in the Supplementary Materials shows the regression between output and targets for each step of the neural network training.

### 4.3. Simulating Future Regional Food Consumption

Food consumption was projected by extrapolating the proportions of each class of each variable in Table 1 based on historical trends. Using the N region as an example, Figure 6 shows how the proportion of each class of variable evolves over time. For the sex variable, the historical shares of the sex of the HRP was used [49]; for schooling and ethnicity, the overall shares of the population were used [68]. For the rest of the regions, Figures S8–S11 of the Supplementary Materials show the same extrapolations.

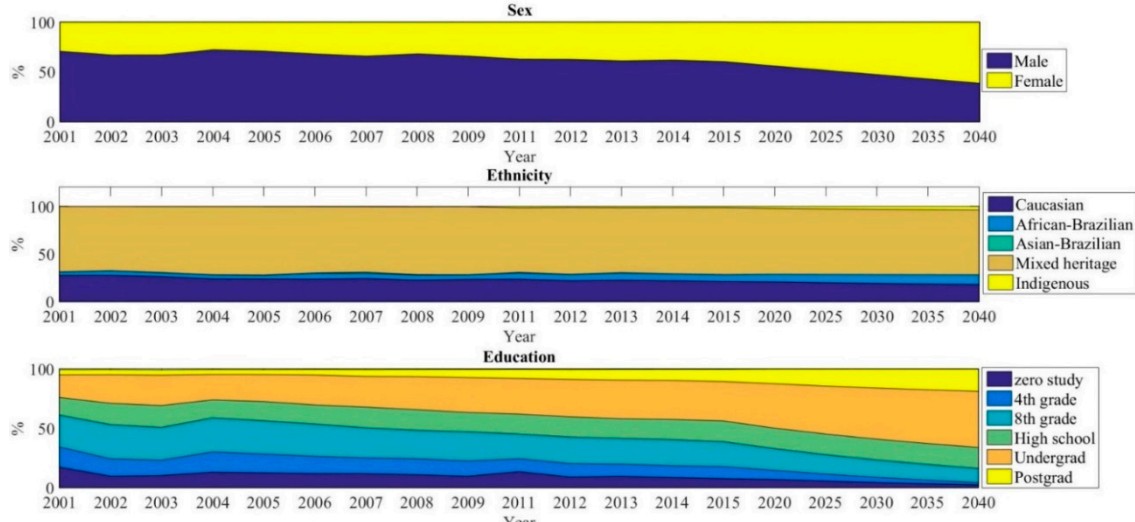

**Figure 6.** Percentage of the population of the North (N) region by sex, ethnicity, and education for the Household Reference Person (HRP), 2001–2040.

The neural networks were run 10,000 times per region, with each run representing a future household, conditioned to the extrapolated variable group shares. For example, in 2040 in the North region, 47% of the HRP will have an undergraduate degree, 61% will be women, and 68% will be mixed-heritage.

After characterizing each household, the first neural network is fed with this information, resulting in the income group to which the household belongs. After that, the second neural network is fed with the same information, along with the resulting group from the first network, yielding the food consumption per capita per household.

## 5. Results

On average, food consumption in Brazil remains roughly constant until 2040, peaking at 4496 kcal/day/capita in 2030 and falling to 4412 kcal/capita/day by 2040 (Table 2 showing signs of convergence between 2020 to 2040 among regions. Some consumption levels appear high; for example, average food consumption in S and CW is close to 5500 kcal/day. It is necessary, however, to keep in mind that consumption is treated from the retail perspective, which is higher than intake as shown by Souza et al. [40].

**Table 2.** Projected average food consumption in Brazil and its regions, 2020 to 2040.

| Kcal/Capita/Day | 2020 | 2025 | 2030 | 2035 | 2040 |
|---|---|---|---|---|---|
| Brazil | 4459.1 | 4493.7 | 4496.1 | 4471.9 | 4412.6 |
| North | 3980.9 | 4031.3 | 4009.9 | 3975.1 | 3966.2 |
| Northeast | 3609.1 | 3713.2 | 3715.4 | 3746.8 | 3768.9 |
| Southeast | 4543.6 | 4564.4 | 4620.1 | 4587.1 | 4510.8 |
| South | 5709.3 | 5652.3 | 5527.5 | 5459.1 | 5274.4 |
| Central-west | 5277.7 | 5287.1 | 5261.2 | 5184.8 | 5094.6 |

Attention is drawn to the fact that the S region presents the highest reduction in average food consumption by 2040. The increase of Afro-Brazilian and mixed heritage households in the region, coupled with the higher level of education expected, culminates in the reduction of average consumption. This can also be seen in the CW. The opposite, however, is expected in the N and NE. With higher education, consumption increases in those regions. Further attention is given to the regional differences in the next sections, divided into sex, ethnicity, education, income, and region.

*5.1. Sex*

The results show that while female-led households tend to consume less food, the male HRP group is more susceptible to changes when it comes to food consumption, which increases at a much faster pace than the opposite group. In 2040, the NE has the highest number of female HRP (72% of households), and food consumption increases from 3609 kcal/capita/day in 2020 to 3769 kcal/capita/day in 2040. Examining each region, inequalities in food consumption based on sex become apparent, with male HRP increasing considerably in all regions except S (Figure 7). NE has a similar pattern to the N region when it comes to female-headed households, which, in 2020, consume 15% less than male-headed ones, and the analysis shows an increase to 26% in 2040.

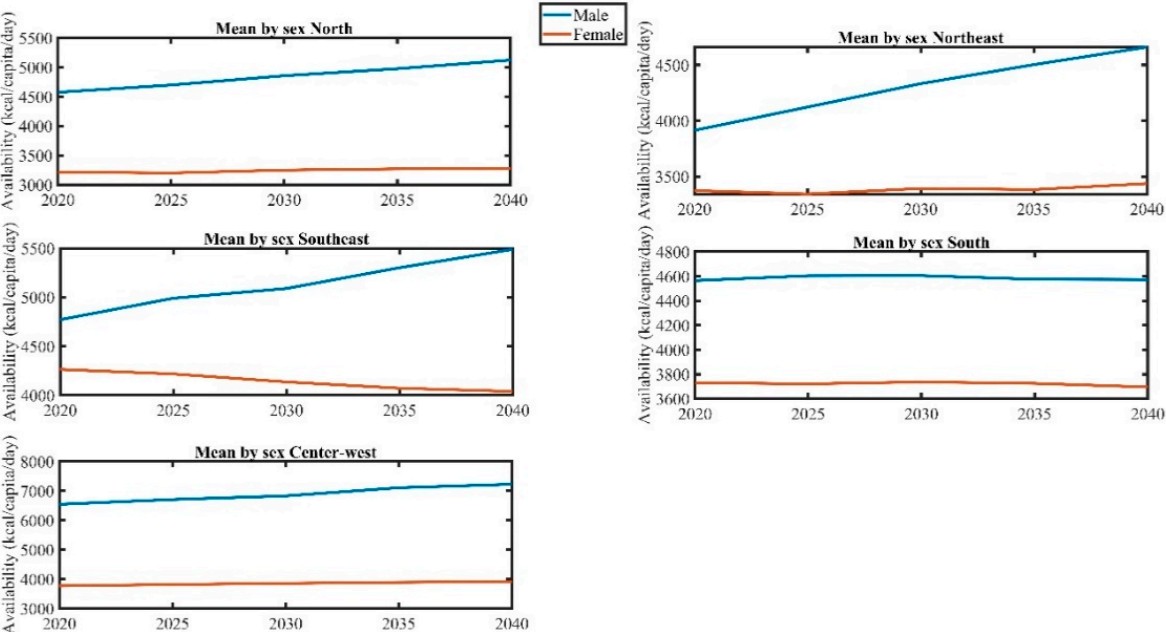

**Figure 7.** Average food consumption evolution in each Brazilian region from 2020 and 2040 by sex group.

When it comes to sex and ethnicity, in 2040, white male dominated households consume, on average, 4405 kcal/day/capita. On the other hand, black female dominated households consume 3431 kcal/capita/day, i.e., 22% less. Nationally, female-headed households will consume 3239 kcal/capita/day compared to 5119 kcal/capita/day by male headed households in 2040, a difference of almost 60%.

*5.2. Ethnicity*

Historically, Afro-Brazilian and mixed heritage households have had lower incomes—even with similar levels of education—reflecting Brazil's inequalities. As a result, food consumption amongst these households is typically lower, and the results show this trend is likely to continue. As the proportion of Afro-Brazilian and mixed heritage households increases, those regions which currently have greater representation of white people (S/SE) see food consumption decrease (Figure 8).

The NE shows, on average, the lowest differences in food consumption among ethnic groups, but inequalities in consumption between Caucasians and Afro-Brazilian increases from 2020 to 2040. The N, on the other hand, is the only region in which Afro-Brazilians have greater food consumption than Caucasians, due to the specific regional characteristics regarding income and access to food by different ethnicities [9,80].

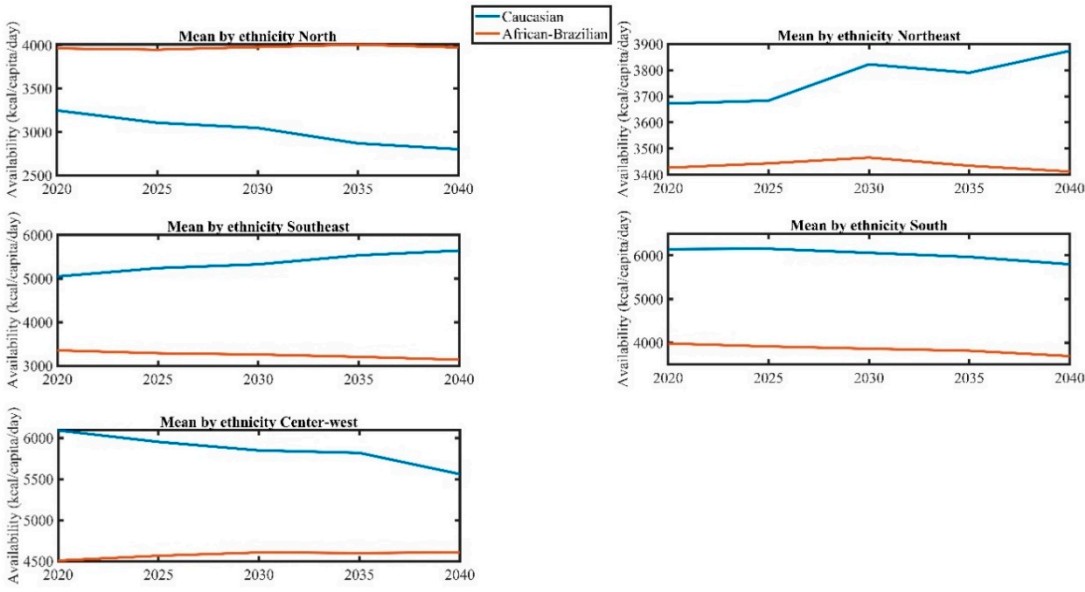

**Figure 8.** Average food consumption evolution in each Brazilian region from 2020 and 2040 by extreme ethnicity group.

## 5.3. Education

Although the N shows an increase in educational levels, with 47% having an undergraduate degree in 2040, the region still has the lowest proportion in Brazil. This is important since higher levels of education tend to lead to an increased income. Most importantly, the increase in educational level in most regions represent a decrease in the average household consumption (Figure 9), consequently reducing also the disparity between the two extreme groups, especially combined with the increase of female and African-Brazilian education.

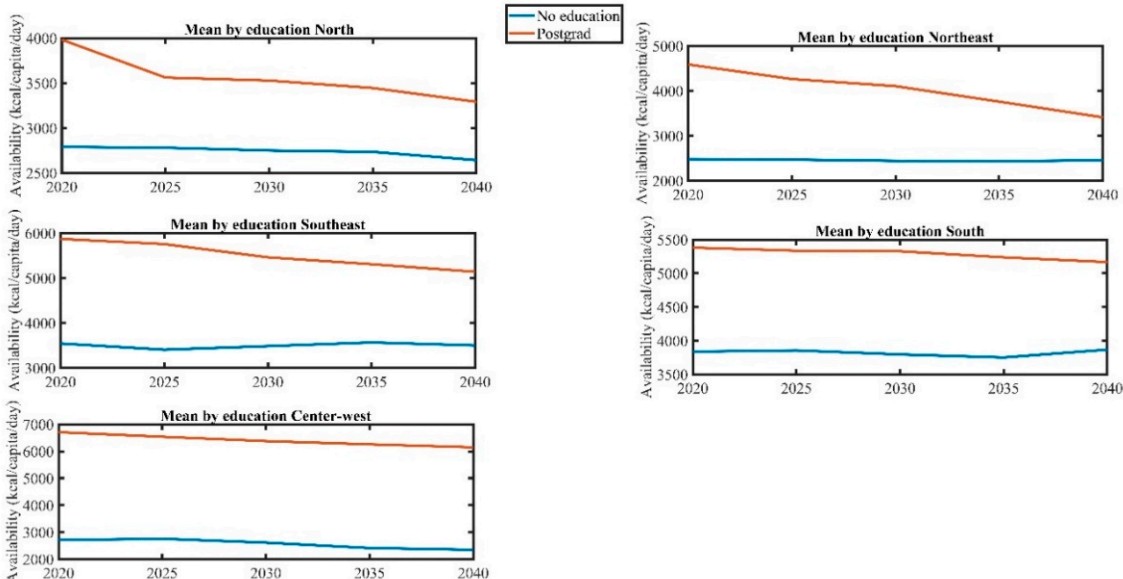

**Figure 9.** Average food consumption evolution in each Brazilian region from 2020 and 2040 by extreme education group.

In fact, this is applicable to all regions and the reduction of food consumption in the 'Postgrad' group is seen across the country. At the other extreme, the 'No education' group also shows reduction in the CW and N regions, which is the most unequal region of the country.

*5.4. Income*

With regard to income, while food consumption in the N remains at 4000 kcal/day between 2020 and 2040, there is an increase in consumption by lower income households and a decrease by higher income households (Figure 10). This is also seen in the NE, as shown in Figure 10.

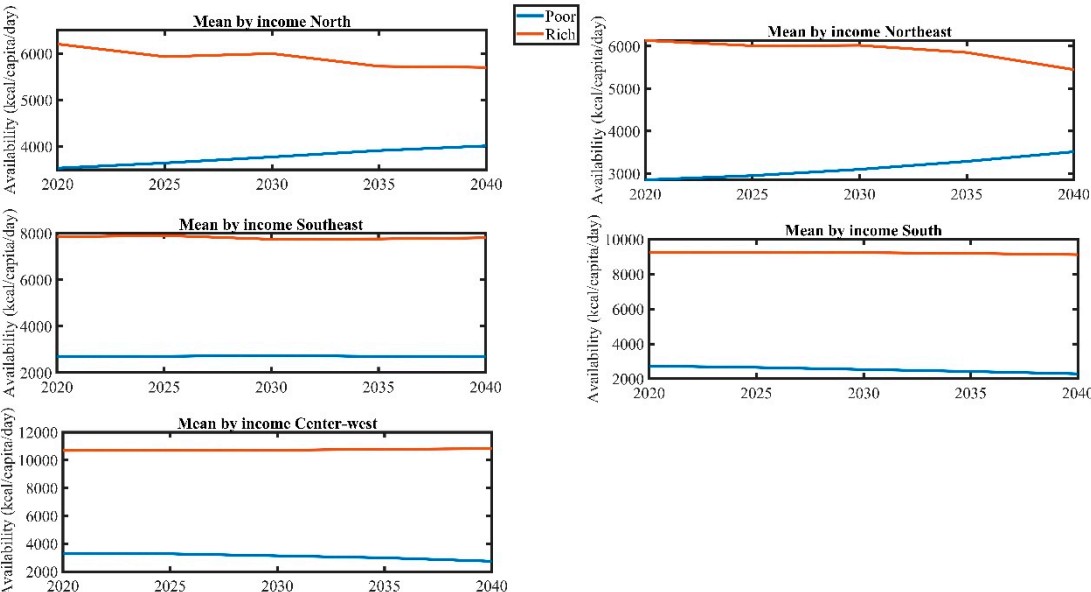

**Figure 10.** Average food consumption evolution in each Brazilian region from 2020 and 2040 by extreme income group.

The SE shows little change in levels of food consumption, although higher income households always have greater consumption than lower income households. The CW and S show a fall in food consumption amongst lower income households, highlighting deepening inequalities between income groups.

The reduction of average household food consumption in time in some regions by the wealthier group is due to the increase in the female headed households in that group coupled with the increase of African-Brazilian and mixed heritage. This means that more people in the rich group will be female and of African-Brazilian descendancy, but these categories consume less than their counterparts in the same income group (Caucasian and male headed households), affecting the final average.

*5.5. Regional Differences*

Figure 11 shows the evolution of food consumption divided into bands of consumption in kcal/day. Each color of the pie charts corresponds to a band, ranging from lower than 2000 kcal/day to more than 7000 kcal/day. From Figure 11, it is evident that households will still be under the recommended average daily food intake of 2000–2500 kcal/day [40,81], most notably in the N region, followed by the NE region. Moreover, there is an increase in the average consumption brackets (from 3000–4000 kcal/day) but also in the higher tier of consumption (>7000 kcal/day), showing the existence of the "double burden of malnutrition".

The SE region has the highest GDP in Brazil; however, the frequency in the band of recommended intake is the highest of them all. Considering that availability is higher than intake, it suggests that many households will continue to be under the intake threshold of 2000 kcal/day. In the S and CW, the highest bands of food consumption have a decrease in frequency, and the average bands have an increase, which result in a slight decrease in inequality. These two regions also present the highest decrease in food consumption, following the already mentioned increase in populations shares of women and African-Brazilian people.

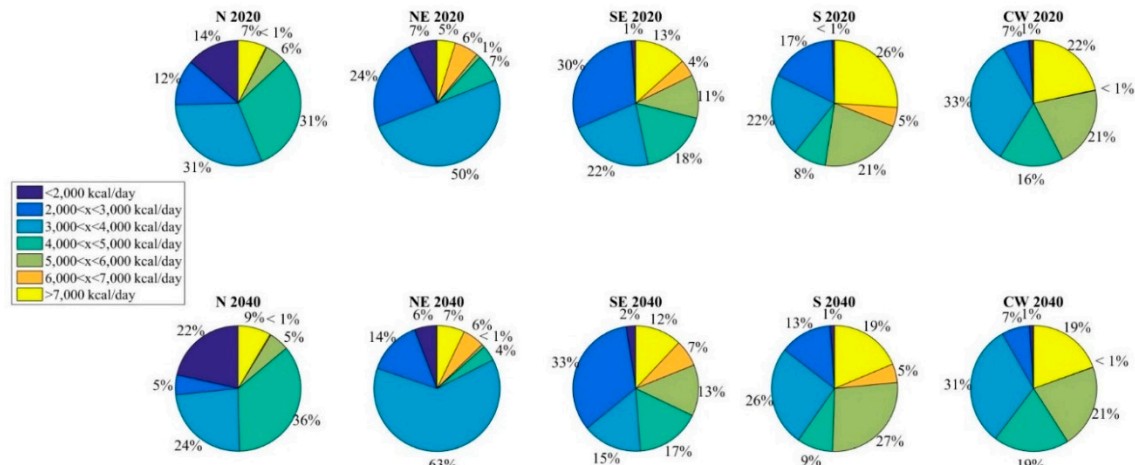

**Figure 11.** Pie chart of household frequency in each Brazilian region for 2020 and 2040 by food consumption bracket.

## 6. Discussion

Food consumption has mostly been analyzed to define the necessary supply from the agricultural sector. In the majority of studies, food consumption is projected in an aggregated fashion dependent on basic macroeconomic such as GDP per capita and population [19,25,82]. Even though other models have attempted to include socio-economic variables, such as income inequality in their analysis [21,32], the focus of those studies have typically been to evaluate the impacts on land use and on the agricultural sector rather than understanding the disparities and inequalities of consumption. Additionally, the interest in sub-national or regional food inequality has been absent from the literature and countries, like Brazil, tend to be aggregated with other countries based on location or socio-economic development [21,27].

The inequalities in Brazil in food consumption, therefore, have never been the focus of a methodological study, leaving important knowledge gaps, such as: 'What are the regional differences in food inequalities/consumption?', What is the nature and extent of food inequalities/consumption?, and, most importantly, 'What are the projected trends in inequality and disparities of consumption?'

This paper has addressed these gaps by developing a methodology to assess food inequalities according to region and disparities among demographic groups. It has used household expenditure data to train neural networks capable of prognosticating future trends in food consumption according to region and demography and to identify regional differences in food consumption and the nature of these differences. In comparison to other models that base their assumptions on traditional theories, such as Engel's curve [25,83] and general equilibrium models [84], this simulator uses modern learning models capable of approximating any function without presuming simplistic relationship between the variables of interest. With this feature, neural networks are capable of learning how dependent and independent variables relate to each other without predefining an analytical structure.

The simulator is sensitive to current household variables and is replicable to other countries/regions, depending on the granularity of data and the availability of HES. Improvements, however, are required in the method to project future consumption, which currently is only based on the evolution of historical trends of household characteristics. The publication of Brazil's 2017 HES would facilitate this process.

The results of this study show that important inequalities remain when it comes to food consumption and that this differs by region and demographics. It finds that the 'poorer' N and NE of Brazil experience the lowest consumption of food and are therefore the most food-vulnerable regions. This trend continues out to 2040. The 'richer' S and SE have higher food consumption, but this varies according to sex, ethnicity, education, and income.

There is also evidence that education and income can secure food consumption and are useful to reduce the inequalities and disparities in regions, including the N, NE, and SE, but certain

socio-economic groups within those regions still consume less than the necessary amount of calories per day. On the other hand, certain groups increase in consumption to amounts beyond the recommended healthy intake [36,85], including socio-economic groups from the lowest strata of income, even if in less frequency. These two complete opposite groups show the disparities within the country, highlighting that is important to not only focus on food access to the most vulnerable but also to establish measures to reduce excessive food consumption, as established by the Brazilian FNS [6,70].

As discussed in the section on Brazilian food policies, the country had laws, policies, and frameworks in place to deal with inequalities, including food and nutritional inequalities. However, five years after the failed attempt to relaunch the FNS, not only is it necessary to deal with the food access by the most vulnerable due to the latest economic declines [1] but also to deal with excess consumption, the so-called "double burden of malnutrition", where undernutrition and obesity co-exist [6,23]. The "double burden" brings serious macro effects on health [36,60,63], the economy [24,86], and the environment [87–89]. Therefore, beside the measures existing in the FNS, it is also important to tackle other contextual inequalities shown that this study has shown has a direct influence on food security, including gender disparities, income inequality, education, and regional differences. Furthermore, challenges such as a growing population, climate change, and pandemics [15,16,90], will continue to place pressure on socio-economic systems, ultimately affecting food security and hunger.

The SDGs provide a global agenda for sustainable development, and tackling inequalities is a vital underpinning concern. To combat inequalities in food consumption, the SDGs provide a basis for national guidelines. Even though food is directly addressed in SDG2 [8], results from this study show that the issue also has direct link with SDG 1 (end poverty), SDG 3 (health and wellbeing), SDG 4 (education), SDG 5 (gender), SDG 10 (reduced inequalities), and SDG 12 (responsible consumption and production).

As the results show, income and poverty are linked with disparities in food consumption and food insecurity, corroborated by many other studies indicating the food insecurity of the most poor [91–95]. SDG 1 focuses on ending poverty, which has an important role in tackling inequalities in food consumption [7], especially by ensuring, particularly for the poor and the vulnerable, equal access to public services and rights to ownership of land and other forms of property (Target 1.4) and by creating solid policies, or guaranteeing the continuation of policies, at all jurisdiction levels based on pro-poor and gender-sensitive development strategies to accelerate poverty eradication (Target 1.B).

Education is another important determinant of food consumption, as seen in this study and in other publications [96–98]. Addressing the low levels of education is required to support food access and to tackle the increase in excess food consumption. SDG 4 has a comprehensive set of targets to make education more inclusive, eliminate disparities and ensure equal access to all levels of education and vocational training for the vulnerable.

The relationship between income inequality and food is fundamental, with strong moral and societal value [21,43]. To address income inequality, SDG 10 aims to achieve income growth of the bottom 40% of the population at a rate higher than the national average (Target 10.1), and encompasses other inequalities by targeting the socio-economic empowerment and political inclusion of all (Target 10.2). Moreover, gender inequalities need to be treated specially with the expectation of increase of consumption in male households, as seen in the results of this study. Female-headed households require particular attention to avoid food insecurity [96,99], which requires the systematic inequalities to be tackled. Recognizing and valuing unpaid care and domestic work, social protection policies, the promotion of shared responsibility within the household (Target 5.4 [8]) and ensuring women's full participation and equal opportunities for leadership at any level of decision-making (Target 5.5 [8]), should be considered as indirect measures to guarantee sustainable food security.

On health, SDG 3 targets a one-third reduction of premature mortality from non-communicable diseases (which are to some extent connected to excess food consumption [5]) through prevention and treatment (Target 3.4). Moreover, food consumption must be managed to increase quality [5,85].

SDG 12 brings targets to deal with that issue by ensuring that people, regardless of their demographic group or region, are aware of more sustainable life styles and in harmony with nature (Target 12.8).

As the world recovers from the Covid-19 pandemic, it is vital that efforts to deliver the SDGs are accelerated, with particular attention paid to actions that may avoid the likely increase in poverty [14]. The increase in poverty can have a direct impact not only on personal income but also access to education. As seen in the results of this study, higher education levels imply higher food security. However, there is little evidence that, especially with the Covid-19 pandemic, the Brazilian government will be able to continue to fund higher education and maintain the increase in schooling delivered during the previous decade, which will likely have negative outcomes for food access, with particular impacts on the most vulnerable.

## 7. Conclusions

This research presents a first step to move models away from aggregated macroeconomic variables to address food consumption at the household level. Now, more than ever, models like the one developed in this study are necessary to address individual behavior and to understand the evolution of inequalities within a country or region.

Results have shown that policies are required not only to guarantee access to food to the poorest but also to tackle the increase in consumption by the richest. Such policies would support the delivery of SDGs 2, 3, and 12. Furthermore, this research has demonstrated that inequalities in food consumption continue and will continue to exist in Brazil if income, gender, and regional inequalities are not addressed. Even though the country was doing well in achieving the SDGs, the last five years have seen a reduction in the level of governmental commitment to social investments, and there is little evidence that the country will be able to reach SDG related to inequality, such as SDG 5 and SDG 10, let alone SDG 2.

The use of neural networks allowed the analysis of food consumption on the regional level and the identification of the several facets of Brazilian reality. However, there is room for improvement in data inputs, especially to determine the evolution over time of food consumption. Finally, future research should focus on the application of this methodology in other countries and cross-compare results to understand if the differences within countries are also applied to differences between countries and to understand if the nature of inequality varies, as well.

**Supplementary Materials:** The following are available online at http://www.mdpi.com/2071-1050/12/15/6132/s1, Figure S1: Layout of the 2002–2003 Brazilian Household Expenditure Survey, Figure S2: Monthly personal income (R$/capita/household in R$ of 2002) in 2002 and in 2008, Figure S3: Percent of households per type of household, ownership, type of floor and sewage connection, Figure S4: Average kilocalorie consumption per capita (kcal/capita) from 1970 to 2013, Figure S5: Resulting clusters divided by region, crossing income and consumption, Figure S6: Confusion matrix for the best neural network, Figure S7: Regression between output and targets for each step of the neural network training, Figure S8: Shares extrapolation of Schooling, sex and color for the person of reference of households until 2040 for the Northeast region, Figure S9: Shares extrapolation of Schooling, sex and color for the person of reference of households until 2040 for the Southeast region, Figure S10: Shares extrapolation of Schooling, sex and color for the person of reference of households until 2040 for the South region, Figure S11: Shares extrapolation of Schooling, sex and color for the person of reference of households until 2040 for the Center-west region, Table S1: Average household income, inhabitants and personal income in 2002, Table S2: Average household income, inhabitants and personal income in 2008, Table S3: Life conditions, income and share of each type of household in 2002, Table S4: Life conditions, income and share of each type of household in 2008, Table S5: Average individual income by group of people and their share of total population in 2002, Table S6: Average individual income by group of people and their share of total population in 2008, Table S7: Resulting regional consumption in 2002, Table S8: Resulting regional consumption in 2008.

**Author Contributions:** Conceptualization, P.G.M., J.T. and C.d.O.R.; methodology, P.G.M. and C.d.O.R.; validation, J.T. and C.d.O.R.; investigation, P.G.M.; resources, A.H.; data curation, P.G.M.; writing—original draft preparation, P.G.M. and J.T.; writing—review and editing, P.G.M., J.T., C.d.O.R. and A.H.; visualization, P.G.M.; supervision, C.d.O.R.; project administration, C.d.O.R.; funding acquisition, P.G.M., C.d.O.R. and A.H. All authors have read and agreed to the published version of the manuscript.

**Funding:** This research was funded by FAPESP, grant number 2014/50279-4 and C.d.O.R. was funded by CNPq grant number 307126/2018-8.

**Conflicts of Interest:** The authors declare no conflict of interest.

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
