# Peer review of "A Simulator to Determine the Evolution of Disparities in Food Consumption between Socio-Economic Groups: A Brazilian Case Study"

_sustainability, doi:10.3390/su12156132_

Round 1
Reviewer 1 Report
I see a lot of improvement in the new version of the article, but it still needs to be corrected.
I don't like how the authors responded to the review. I suggested them a language correction and they said it was difficult to do it 'without pointing specifically to what language and style errors I was referring'. Dear Authors, it is not the task of the reviewer to be a language corrector/editor and to indicate all grammatical/style errors. If I suggest the need for English proofreading, then you should return the text to a professional service dealing with it.
Your answer to comment 16 is not satisfying either. Figure 7 is illegible. The values cannot be read. I do not suggest giving an additional table, but replace Figure 7 with the table.
Figure 12 is unacceptable. It's completely unclear. You must think about another form of presenting these results. Do not expect me to tell you which one because I am not the author and I have no insight into the data. Maybe a few separate drawings? I don't know, but it can't stay that way.
Line 25 - the word 'vulnerable' refers to what?
You need to change the keyword: food availability into food consumption
Author Response
Comment: I don't like how the authors responded to the review. I suggested them a language correction and they said it was difficult to do it 'without pointing specifically to what language and style errors I was referring'. Dear Authors, it is not the task of the reviewer to be a language corrector/editor and to indicate all grammatical/style errors. If I suggest the need for English proofreading, then you should return the text to a professional service dealing with it.
Answer: We do apologize if the way we responded to your comment has offended you in any way, however, it did not by any mean meant that we did not take your comment into consideration. We actually performed a thorough review, which includes reviews from 2 of our co-authors that are native English speakers. To show you that we have done the review you asked, we are submitting the previous version of the manuscript with the tracked-changes. When we asked you to be more specific about the language and style errors, we meant if you could point out to a specific error that you might have come across. We agree with you that this is not the job of the reviewer to proofread the text for English errors, but it would have been of great help if you had managed to be more specific. Either way, we have done the revision asked. Hopefully the quality of the writing is within your standards.
Comment: Your answer to comment 16 is not satisfying either.
Answer: We have changed the figure 7 to a table, as you proposed in comment 16.
Comment: Figure 7 is illegible. The values cannot be read. I do not suggest giving an additional table, but replace Figure 7 with the table.
Answer: Although we disagree, we do not wish to delay any further the review process and we have changed figure 7 for a table, called table 2.
Comment: Figure 12 is unacceptable. It's completely unclear. You must think about another form of presenting these results. Do not expect me to tell you which one because I am not the author and I have no insight into the data. Maybe a few separate drawings? I don't know, but it can't stay that way.
Answer: Thank you for your comment. We have changed it into pie charts only for years 2020 and 2040 to show the distribution of households within each consumption bracket to facilitate visualization. Indeed it improved the understanding of the figure and we hope this is enough to satisfy your comment.
Comment: Line 25 - the word 'vulnerable' refers to what?
Answer: We have added an explanation to the word “vulnerable” in its first instance in line 72, with an addition reference to the use of the term “vulnerable”.
Comment: You need to change the keyword: food availability into food consumption
Answer: Thank you, the keyword has been changed.
Reviewer 2 Report
The paper is appropriate for publication in my opinion.
Author Response
Thank you, we appreciate the review and the compliments.
Reviewer 3 Report
Motivation
This resubmitted paper has been much improved. I just have one comment to better structure the motivation stated in the “Introduction”. It is stated that “Now, the country not only has to deal with the food access by the most vulnerable due to the latest economic declines, but also to deal with excess consumption, the so called “double burden of malnutrition”. However, the critical issues concerns “double burden of malnutrition” was not mentioned in the “Introduction”. According to WHO, the term “double burden of malnutrition” concerns “The double burden of malnutrition is characterised by the coexistence of undernutrition along with overweight and obesity, or diet-related noncommunicable diseases, within individuals, households and populations, and across the lifecourse.” (https://www.who.int/nutrition/double-burden-malnutrition/en/) It is obvious that this phenomenon is prevalent for countries that experience high consumption inequality. Adding the linkage between the present study and the major conclusion made can help to improve the motivation statement.
Method
This study proposes to use a neural network model to identify the evolution of regional food availability over time. To better explain the advantage of the use of neural network approach, the author(s) need to provide an overview of the methods previously used to forecast food consumption. Even it was stated that the neural network approach does not assume a predetermined temporal analytical structure as is in [19]; making contribution through improving over just one of the previous research is far from providing a concrete background of the advantage of the research method adopted in this study.
Result and Discussion
I’d like to suggest removing the sentences regarding food availability in the paper, for example “Food availability has mostly been analyzed from the supply side. In the majority of studies, food availability is projected in an aggregated fashion dependent on basic macroeconomic such as GDP per capita and population [20,23,90]” since food availability is not the focus of the present study.
Author Response
Comment: This resubmitted paper has been much improved. I just have one comment to better structure the motivation stated in the “Introduction”. It is stated that “Now, the country not only has to deal with the food access by the most vulnerable due to the latest economic declines, but also to deal with excess consumption, the so called “double burden of malnutrition”. However, the critical issues concerns “double burden of malnutrition” was not mentioned in the “Introduction”. According to WHO, the term “double burden of malnutrition” concerns “The double burden of malnutrition is characterised by the coexistence of undernutrition along with overweight and obesity, or diet-related noncommunicable diseases, within individuals, households and populations, and across the life course.” (https://www.who.int/nutrition/double-burden-malnutrition/en/) It is obvious that this phenomenon is prevalent for countries that experience high consumption inequality. Adding the linkage between the present study and the major conclusion made can help to improve the motivation statement.
Answer: Thank you for your comment. We have added the concern on “double burden” in the introduction as mentioned.
Comment: This study proposes to use a neural network model to identify the evolution of regional food availability over time. To better explain the advantage of the use of neural network approach, the author(s) need to provide an overview of the methods previously used to forecast food consumption. Even it was stated that the neural network approach does not assume a predetermined temporal analytical structure as is in [19]; making contribution through improving over just one of the previous research is far from providing a concrete background of the advantage of the research method adopted in this study.
Answer: Thank your comment. However, we have already added a large new section called “previous models” (section 2.1) in the new literature review. It is a 628-words section that talks about previous models, types of previous models and also cites a review done by Flies et al., who reviewed previous models for food consumption projection. We hope this was enough to clarify the advantages of our method.
Comment: I’d like to suggest removing the sentences regarding food availability in the paper, for example “Food availability has mostly been analyzed from the supply side. In the majority of studies, food availability is projected in an aggregated fashion dependent on basic macroeconomic such as GDP per capita and population [20,23,90]” since food availability is not the focus of the present study.
Answer: Thank you for your comment. We have removed the sentences regarding food availability mentioned.
This manuscript is a resubmission of an earlier submission. The following is a list of the peer review reports and author responses from that submission.
Round 1
Reviewer 1 Report
I read the article with pleasure. It concerns an important and frequently raised problem, but still, it is very original. An interesting research approach was used, which gave surprising results. I also appreciate the practical dimension and the fact that the method can be used in other countries.
I have three main points:
1. I noticed grammar, language and style errors, so language proof is necessary. This comment refers especially to the Discussion section.
2. There is a technical problem with references to tables and figures in the whole text. Instead of reference, there is a message "Error! Reference source not found".
3. The Discussion section should be rethought. There is a weak reference to existing literature and one cannot see your contribution clear enough.
Detailed comments:
Line 95 - "The N and NE regions presents the highest income inequalities" - it is not clear whether this statement refers to inequality within the region or within the whole country. Besides, there is a grammar mistake in this sentence - present not presents.
Line 106 - I see no point in distinguishing section 2.2. It is too short and does not raise the issue of food availability patterns as stated in the title. The food availability patterns are rather described in section 2.3
Line 166 - I would rather change the title of section 2.4. It describes recent food policies in Brasil and the issue of food and nutrition security is rather marginal.
There is no reference to Figure 3 in the text.
Line 226 and 229 - food availability is a "dependent" variable here
Line 246 - there is a technical issue with the equation - question marks. Besides, what is the point in presenting the equation twice?
Line 256 - what is the point in presenting the equation twice?
Line 259 - "of the person" is redundant
Line 271 - what is section 0
Page - there is a consistency problem with equation numbers and the reference numbers in the text
Line 320 - rather Tables S7 and S8
Line 350-351 - rather Figures S8 to S11
Figure 7 - shows nicely differences between region, but a table with more detailed data would be useful to see the development within a single regions.
Explain what is the reason for the strong decrease in food availability in the South.
Line 375 - "the male HRP group is more susceptive to income and education changes when it comes to food availability" - it does not result from the research. The relationship between education and food availability has not been shown here.
Line 406 - "The N, on the other hand, is the only region in which Afro-Brazilians have greater food availability than Caucasians" - explain why?
Line 448 - where the figure 13?
Figure 12 is unclear and the title is wrong, i.e. it is not by extreme education group. Please present the data in a different form.
Lines 496-499 - the sentence is unclear and with typo
Line 503 - "attempt to relaunch the , not only" - something is missing here
Line 525-528 - do not use the word "also" three times in one sentence
Reviewer 2 Report
The paper is very interesting. It presents a hot topic that is very relevant with SDGs. The authors managed to present explicitly the problem, the methodology used and the results developing a simulator to analyze food availability from the demand side taking into consideration plenty of attributes (region, sex, ethnicity, education and income). The most interesting part is the combination of all these attributes and the interpretation of the results. The idea to use both FFQ and FIA approach is very cleaver and innovative. Literature review is updated and helps reader to understand the Brazilian way of living, how it has changed the recent years as well as the reasoning beyond these changes. Th language had no errors.
It just needs a very careful reading in order to correct some minor editing mistakes (e.g in some places there appears "Error! Reference source not found)".
Overall, I enjoyed it a lot and I suggest publication in present form (after correcting some very minor editing errors).
Reviewer 3 Report
Motivation
There is a major flaw of design in this study. The main focus of this study is to analyze food availability from the demand side and estimate the evolution of inequality in food availability using household expenditure surveys. I do not have any problems with investigating expenditure inequality from the demand side or using the household expenditure data. However, it is quite awkward to talk about food availability from the demand side since food availability is totally dependent on the supply side factors instead of the demand side. This is similar to the delineation of the food consumption behavior in examining the impacts of supply shock in the market.
Method
This study proposes to use a neural network model to identify the evolution of regional food availability over time. The author(s) did not provide an overview of the methods previously used to forecast food availability. Moreover, what are the advantages of a neural network model over the methods used in the past were not mentioned either.
Model Specification
Instead of basing the prediction of food availability on the supply side factors, this study uses household characteristics to simulate food availability based on sex, ethnicity, income and education. Therefore, the prediction made in this study does not concern food availability but rather food consumption.
Result and Discussion
The term “inequality” is misused in this study. In a lot of places, the author(s) talked about “inequality” between the two groups in discussing the results. Inequality is a measure among the population or the sub-population. Since the discussion centers on the comparison in food availability between groups, the author(s) should just say so instead of using the term inequality.